# Site-Specific RNA Editing of Stop Mutations in the CFTR mRNA of Human Bronchial Cultured Cells

**DOI:** 10.3390/ijms241310940

**Published:** 2023-06-30

**Authors:** Roberta F. Chiavetta, Simona Titoli, Viviana Barra, Patrizia Cancemi, Raffaella Melfi, Aldo Di Leonardo

**Affiliations:** 1Department of Biological, Chemical and Pharmaceutical Sciences and Technologies, University of Palermo, 90128 Palermo, Italy; chiavetta.roberta@gmail.com (R.F.C.); simona.titoli@unipa.it (S.T.); viviana.barra@unipa.it (V.B.); patrizia.cancemi@unipa.it (P.C.); raffaella.melfi@unipa.it (R.M.); 2Centro di Oncobiologia Sperimentale (C.O.B.S.), Viale Delle Scienze, 90128 Palermo, Italy

**Keywords:** cystic fibrosis, premature termination codons (PTCs), RNA editing, CRISPR/dCas13, REPAIRv2, mxABE

## Abstract

It is reported that about 10% of cystic fibrosis (CF) patients worldwide have nonsense (stop) mutations in the CFTR gene, which cause the premature termination of CFTR protein synthesis, leading to a truncated and non-functional protein. To address this issue, we investigated the possibility of rescuing the *CFTR* nonsense mutation (UGA) by sequence-specific RNA editing in CFTR mutant CFF-16HBEge, W1282X, and G542X human bronchial cells. We used two different base editor tools that take advantage of ADAR enzymes (*adenosine deaminase acting on RNA*) to edit adenosine to inosine (A-to-I) within the mRNA: the REPAIRv2 (*RNA Editing for Programmable A to I Replacement, version 2*) and the minixABE (*A to I Base Editor*). Immunofluorescence experiments show that both approaches were able to recover the CFTR protein in the CFTR mutant cells. In addition, RT-qPCR confirmed the rescue of the CFTR full transcript. These findings suggest that site-specific RNA editing may efficiently correct the UGA premature stop codon in the CFTR transcript in CFF-16HBEge, W1282X, and G542X cells. Thus, this approach, which is safer than acting directly on the mutated DNA, opens up new therapeutic possibilities for CF patients with nonsense mutations.

## 1. Introduction

Cystic fibrosis (CF) is an autosomal recessive multiorgan disease that affects the function of exocrine glands, caused by mutations in the transmembrane conductance regulator (CFTR) gene [1]. The CFTR protein regulates chloride secretion and its malfunctioning leads to increased sodium (Na^2+^) adsorption and water influx through the epithelial layers of various tissues such as the pulmonary airways, pancreas, and intestine [2]. This imbalance causes the accumulation of thick fluids and obstructions of ducts that can lead to inflammation and severe infections, mainly in the lung. Various mutations affecting the CFTR gene are responsible for cystic fibrosis and approximately 10% of CF patients worldwide have stop mutations in the protein-coding gene sequence [3]. These stop mutations introduce a premature termination codon (PTC) in the CFTR mRNA resulting in the early termination of translation and the production of a truncated, non-functional CFTR protein [4,5,6]. Currently, there is no effective treatment for CF patients with stop mutations. Nonetheless, in a precision medicine era, novel tools designed ad hoc to restore the CFTR protein are worth investigating to increase the therapeutic opportunities of CF patients carrying stop mutations. To this aim, programmable RNA sequence-specific base editing (hereafter RNA editing) exploiting adenosine deaminases acting on RNA (ADAR 1, ADAR 2), is a valuable approach to correct stop mutations in the CFTR mRNA [7,8,9,10,11]. ADARs are a family of enzymes that catalyze the deamination of adenosine into inosine (A-to-I) in perfectly or partially double stranded RNAs [12,13,14]. Therefore, by modulating the activity of ADARs, it is possible to edit adenosine into inosine [7,10,15], and restore a coding codon that allows the translation process to continue. This strategy is of particular value for UGA stop codons that cause nonsense mutations [12]. Indeed, the inosine is read as guanosine by the ribosome and the amino acid tryptophane is inserted in the newly synthesized polypeptide [12,16]. Moreover, the sequence-specific RNA editing offers a promising and safer approach for correcting CFTR stop mutations compared to DNA editing methods [17,18]. In fact, RNA editing modifies the RNA transcript of a gene rather than the underlying DNA sequence and any side effects would then be less severe and would not be permanent. Recently, two A-to-I conversion systems, named REPAIRv2 (*RNA Editing for Programmable A to I Replacement, version 2*) [19] and mxABE (*mini-dCas13X-mediated RNA adenine base editor*) [20] have been developed to perform RNA editing by targeting the adenosine of the premature UGA stop codon. The REPAIRv2 is based on the catalytically inactive Cas13b protein (*dCas13b*) fused with the adenosine deaminase domain (DD) of the ADAR2 enzyme (h*ADAR2_DD_*. The mxABE system is a compact version of the previous system which uses a truncated and catalytically inactive Cas13X protein (*dCas13X.1)* fused with h*ADAR2_DD_*. In both systems, to recruit the Cas/h*ADAR2_DD_* to the target, a complementary guide RNA (gRNA) is needed. The gRNA is composed of a specific sequence complementary to the mRNA region surrounding the target adenine (spacer) fused to a direct repeat that folds to originate the recognition hairpin for dCas13 protein (scaffold) [19,20]. In the REPAIRv2 system, the two functional elements, dCas13b/h*ADAR2_DD_* and the gRNA expression cassettes, are coded by two separate vectors [19], while in the mxABE system both *dCas13X.1*-h*ADAR2_DD_* and the gRNA are cloned in a single plasmid. The small size of mxABE allows its packaging into AAV particles which might be exploited for in vivo targeted delivery [20]. Here, we show that both approaches successfully recovered the CFTR protein in the human bronchial CFF-16HBEge cells isogenic for the W1282X (c.3846G>A) or the G542X (c.1624G>T) stop mutations (hereafter 16HBE^W1282X^ and 16HBE^G542X^) [21]. Sequence-specific RNA strategies represent a promising frontier in the search for a cure for cystic fibrosis caused by nonsense mutations. These innovative approaches offer the potential to correct CFTR gene mutations and restore the functional CFTR protein.

## 2. Results

### 2.1. CFTR Protein Rescue in the 16HBE ^W1282X^ Cell Using the REPAIRv2 System

Previously, we showed that the REPAIRv2 system could recover CFTR expression in IB3.1 airway epithelial cells (CFTR F508del/W12382X) that are null for detectable CFTR protein, and in FRT-CFTR^W1282X^ engineered cells [22]. Here, we investigated whether the REPAIRv2 base editor is also able to recover CFTR full-length expression in a different cell model of cystic fibrosis, like the human bronchial 16HBE^W1282X^ cells. The REPAIRv2 system uses two different plasmids expressing dCas13b/hADAR2_DD_ and the gRNA with the C:A mismatch, necessary to drive the ADAR2_DD_ to the specific mutant adenine (c.3846 G>A), in different positions with respect to the spacer 3′ end (Figure 1). Indeed, it is widely demonstrated that the position of the mismatch relative to the 3′ or 5′ end of the fragment is important to increase deamination efficiency [19,20].

Human 16HBE^W1282X^ cells were co-transfected with the dCas13b/hADAR2_DD_ expressing plasmid and the plasmid coding for two different gRNAs, CFTR-^W1282X^ 50–32 and CFTR-^W1282X^ 50–34 (50 bps long spacers with C:A mismatch in position 32 or 34). Forty-eight hours after transfection, cells were fixed and visualized using fluorescence microscopy for CFTR expression. The results showed that the transfection with the specific guide RNAs CFTR-^W1282X^ 50–32 + 50–34 was able to increase the percentage of CFTR-positive 16HBE^W1282X^ cells with respect to those untransfected (Unt) or transfected with a non-targeting gRNA (NT) (Figure 2). G418, known as a translational readthrough inducing drug (TRID), was used as a positive control [23].

### 2.2. W1282X- and G542X-Specific gRNA Design and Cloning in the mxABE Platform

Recently, a more compact REPAIRv2-related tool termed minixABE has been reported [20]. This tool, hereafter referred as minixABE or mxABE, relies on a shorter version (mini) of the catalytically inactive dCas13X.1. The main advantage of this system is the presence of both *g*RNA and mini dCas13X.1/hADAR2_DD_ expressing cassettes in a single vector, which also harbors the eGFP expressing cassette. The double BbsI restriction site is present upstream of the direct repeat, coding the scaffold RNA necessary for ADAR2_DD_ recruiting, and was used for the directional cloning of spacer fragments and coding for 50 nts complementary to the specific mutant regions of the CFTR (W1282X or G542X) transcript. Spacers were designed so as to place the C:A mismatch with target mRNA that is specifically recognized by the ADAR2_DD_, in position 25, shown to be the most effective by Xu and collaborators [7] (Appendix A). We then generated the clones *mxABE-CFW1282X 50-25, mxABE-CFG542X 50-25,* and the control clone *mxABE-NT*, where the gRNA is a non-targeting sequence (Appendix A).

### 2.3. Evaluation Using Fluorescence Microscopy of CFTR Protein Recovery before and after Cell Sorting

To assess the efficiency of the mini xABE system in 16HBE^W1282X^ and 16HBE^G542X^ cells transfected with the mxABE-NT control plasmids or with the mxABE-CF^W1282X^ 50–25 and mxABE-CF^G542X^ 50–25 plasmids carrying the specific guide RNAs (*gRNA 50–25*), the CFTR protein recovery was evaluated using immunofluorescence analysis (Figure 3 and Figure 4).

Our findings suggest that the expression of the specific gRNA allowed the recovery of the CFTR protein in the 35% of 16HBE ^W1282X^ cells and in 25% of 16HBE ^G542X^ (Figure 3 and Figure 4). In addition, the number of CFTR-positive cells after the mxABE transfection showed a slight increase in respect to that obtained with the REPAIRv2 tool for the CFTR^W1282X^ mutation (Figure 2 and Figure 3). The efficiency of the mxABE system was higher for the CFTR^G542X^ mutation than for the CFTR^W1282X^ mutation, and the recovery of the CFTR protein with mxABE-CF^G542X^ overcame the one obtained with the G418 treatment (Figure 3 and Figure 4) as showed by IF.

However, the percentage of rescued CFTR cells was not as high as expected. Thus, in the attempt to increase the number of cells to be analyzed for RNA editing with the mxABE, we decided to take advantage of the eGFP-reporter gene carried by the mxABE plasmid to sort the successfully transfected cells.

To this end the 16HBE^W1282X^ and 16HBE^G542X^ eGFP-positive cells were sorted using cytofluorimetry 72 h after transfection with the mxABE vector (Appendix A). Through cytofluorimetry analysis, we found that about 25–30% of 16HBE^W1282X^ and 15–20% of 16HBE^G542X^ cells were eGFP-positive after transfection (Appendix A) mirroring the transfection efficiency. Sorted eGFP-positive cells were then seeded and an immunofluorescence assay was performed. The results showed that 41% (16HBE ^W1282X^) and the 42% (16HBE^G542X^) of the sorted cells were CFTR-positive (Figure 5, Figure 6 and Appendix A). This finding suggests that about 40% of cells were efficiently transfected with the mxABE-system-rescued CFTR protein, regardless of the mutation context, demonstrating the efficacy of the system.

### 2.4. Evaluation of CFTR mRNA Levels Using RT-qPCR

One of the main concerns when dealing with mRNA harboring a premature termination codon as a target for a therapeutic approach is that these mutant transcripts mainly undergo the mRNA nonsense-mediated decay (NMD) surveillance pathway, which is triggered when ribosomes stall on an mRNA harboring a PTC, leading to early degradation [24]. To evaluate the effect of mxABE vector transfection on the levels of CFTR mRNA in 16HBE^W1282X^ and 16HBE^G542X^ cells, we completed RT-qPCR assays on the RNA extracted from treated and control cells. The mRNA levels of CFTR increased in both the mutated cell lines following transfection of the mxABE vector (Figure 7A). In the attempt to obtain a more accurate estimation of the efficiency of the mxABE system, we sorted the cells after mxABE transfection to enrich the population harboring the plasmid. Figure 7 shows that, in both cell lines, the mRNA levels increased after the transfection, revealing a positive effect of the mxABE.

## 3. Discussion

Translational readthrough inducing drugs (TRIDs) are considered a potential therapeutic strategy for treating hereditary diseases caused by stop mutations such as cystic fibrosis. Currently, promising TRIDs to correct PTCs are under investigation [25,26,27,28,29,30,31], but specific and successful treatments against stop mutations are still lacking. In this context, gene therapy approaches are considered as potential effective alleys in restoring the premature stop codons of the mRNA. In particular, mRNA editing tools, like the REPAIRv2 and its more compact version mxABE, have been reported to also be useful for the correction of PTCs because of their precise action on the adenosine of PTCs. Both systems can convert adenosine into inosine (A > I) through deamination that is mediated by the ADAR2 Deaminase Domain (ADAR2_DD_), specifically directed by a guide RNA to the RNA sequence containing the PTC. We investigated the efficacy of sequence-specific RNA editing technologies to correct the premature stop codon (UGA) in the CFTR transcript of isogenic CFF16HBEge human bronchial cells harboring the CFTR ^W1282X^ and ^G542X^ stop mutations. Our results show that the REPAIRv2 system successfully recovered the CFTR protein in 16HBE^W1282X^ cells. In addition, the mxABE tool seems to rescue CFTR protein in both 16HBE^W1282X^ and 16HBE^G542X.^ as shown by IF. Even though the number of CFTR-positive cells was lower than expected, their quantification suggests a CFTR recovery similar to the one observed in the cells treated with G418, a well-known TRID. The restoration, even though partial, of the CFTR protein would suggest the occurrence of specific mRNA editing of the UGA premature termination codon in these cells. In addition, rescued CFTR in some cells appears to be localized onto the plasma membrane, even though it remains to be determined if the protein’s channel function was recovered.

We think that the scored low number of CFTR-positive cells mirrors both the low transfection efficiency and the low level of nonsense transcript present in the cells being edited. Indeed, transcripts with nonsense mutations could be degraded by the NMD pathway, protecting cells from the translation of the mutant transcript.

To evaluate the efficacy of the system bypassing the issue of transfection, we decided to enrich the cell population harboring the mxABE system by sorting cells for eGFP positivity. Cell sorting using cytofluorimetry allowed us to specifically analyze the positive transfected cells with the mxABE, discarding all the untransfected cells. In addition, the RT-qPCR of the sorted cells revealed increased levels of the CFTR transcript, as frequently observed after RNA editing approaches.

We believe that these encouraging results could pave the way for future therapeutic strategies involving specific RNA editing approaches that can also be suitable for innovative delivery systems. Due to its compact conformation, in fact, the mxABE can easily be cloned into recombinant adeno-associated viral vectors (AAVs). In addition, the transient effect of this system, which is acting on RNA instead of DNA, makes these tools potentially safer and more effective in treating stop mutations and, in turn, cystic fibrosis. One major concern with editing techniques, mainly when ADAR’s overexpression is required, is the risk of off-target activity [32]. However, only 20 detectable off-target edits [19,33] were caused by REPAIRv2 and only a basal level of off-target events were caused by the mxABE platform [20], providing high specificity. Moreover, combining the RNA editing tools with other molecules that can preserve the mRNA from degradation (for example, by inhibiting the NMD pathway) could lead to better results. Collectively, our results confirm what we observed previously on FRT and IB3.1 cells [22], and extend the ability of the RNA editing technology to rescue the full-length CFTR protein with both the REPAIRv2 and the mxABE tools [19,20].

Finally, sequence-specific RNA strategies represent a promising frontier in the search for a cure for cystic fibrosis caused by nonsense mutations. These innovative approaches offer the potential to correct CFTR gene mutations and restore functional CFTR proteins. Despite this, several challenges remain, including delivery systems and advancements and refinement of gene RNA editing techniques, which hold the potential to revolutionize CF treatment of individuals who cannot benefit from current therapies.

## 4. Materials and Methods

### 4.1. MinixABE Platform: Spacers Design and Cloning

To generate the spacer double-stranded DNA fragments to be cloned in the U6-BbsI-DR-CMV-miniCas13X.1-Repairv2-BGHvector (Addgene #171383), specific complementary oligonucleotides corresponding to the 50 bps sequences surrounding the target adenosines were designed (Appendix A). AACG or CAGC 5′ overhangs were introduced in the sequences to ensure directional cloning in the BbsI restriction site of the vector. Complementary oligonucleotides were allowed to pair by heating up to 94 °C and allowing them to cool down to RT in a medium salt buffer. After digestion with BbsI restriction enzyme and agarose gel purification, the mxABE backbone vector was ligated by T4 ligase with each of the DNA fragments (Appendix A) and then transformed into XL1 Blue bacterial competent strain. Positive clones were selected with colony PCR with the U6 forward primer (Appendix A) coupled with the cloned reverse oligonucleotides. From all colonies, we obtained the expected amplicons of 2425 or 2405 bps (Appendix A), and in each case one was randomly selected for further characterization. Plasmid DNA was extracted by a commercial kit (Invitrogen, Thermofisher Scientific, Monza, Italy) and gel verified and sequenced with U6 forward primer (BMR genomics, Padova, Italy).

### 4.2. Vectors and Clones

For the REPAIRv2 system we used the following plasmids:-pC0055-CMV-dPspCas13b-GS-ADAR2DD(E488Q/T375G)-delta-984-1090, from Addgene (RRID: Addgene_103871);-pC0043-PspCas13b crRNA backbone, from Addgene (RRID: Addgene_103854);-CFTR-gRNA W1282X 50–32 and CFTR-gRNA W1282X 50-34, designed by us [22].

For the mxABE system we used the following plasmids:-U6-BbsI-DR_CMV-minidCas13X.1-REPAIRv2-BGHpA_CMV-EGFP-BGHpA, from Addgene (RRID: Addgene_171383);-mxABE-NT, coding for a control gRNA (non-targeting) (Appendix A);-mxABE-CFW1282X 50–25, coding for a gRNA specific for the W1282X mutant region, 50 nts long, with a mismatch in position 25 with respect to 5′end (Appendix A);-mxABE-CFG542X 50–25, coding for a gRNA specific for the G542X mutant region, 50 nts long, with a mismatch in position 25 with respect to 5′end (Appendix A).

### 4.3. Cell Culture Conditions

CFF-16HBE^W1282X^ and CFF-16HBE^G542X^ cells (provided by the Cystic Fibrosis Foundation, Lexington, MA, USA) and 16HBE14o^−^ (parental, used as WT), were grown onto PureCol^®^ (Cell Systems GmbH, Troisdorf, Germany) bovine-collagen-coated flasks in MEM (Minimum Essential Medium; Gibco, Life Technologies, Monza, Italy)) supplemented with 10% fetal bovine serum 100 U/mL penicillin and 100 μg/mL streptomycin (Gibco, Life Technologies, Monza, Italy) in a humidified atmosphere of 5% CO_2_ in air at 37 °C. Cells were split every 3 days using TrypLE Express (Gibco, Life Technologies) and seeded 1:3 for maintenance in T-25 collagen-coated flasks.

### 4.4. Cell Transfection

Cells at 70–80% of confluency were transfected with the appropriate plasmids coding for the different guide RNAs, using Lipofectamine 3000 (Life Technologies, Monza, Italy) or FuGENE 4 K reagent (Promega, Milan, Italy) according to manufacturer’s instructions. In the transfection with the REPAIRv2 the plasmid coding for dCAS13b/ADAR2DD was co-transfected with the ones coding for the two different gRNAs. The transfection mix was incubated for 8 min with Lipofectamine 3000, (Life Technologies Monza, Italy) or for 15 min with FuGENE 4K (Promega, Milan, Italy) at room temperature and left for 72 h after administration.

### 4.5. RNA Extraction, cDNA Synthesis, and RT-qPCR

Total RNA from transfected cells was extracted using the RNeasy Mini or Micro Kit (QIAGEN, Milan, Italy), depending on the number of cells according to the manufacturer’s instructions, and retrotranscribed with the High-Capacity cDNA Reverse Transcription Kit (Applied Biosystems™, ThermoFisher Scientific, Monza, Italy). The cDNAs (50–75 ng/replicate) were then added to a mix containing SYBR^TM^ Green (Applied Biosystems™; ThermoFisher Scientific, Monza, Italy; 12.5 µL/replicate), forward and reverse primers (1 µM each) (Appendix A), and H_2_O for a final volume of 15–20 µL/well. Samples were analyzed with the thermocycler Applied Biosystems 7300 Real Time PCR System (ThermoFisher Scientific, Monza, Italy) (15″ at 95° C, 60″ at 60° C, for 40 cycles).

### 4.6. Immunofluorescence

Cells were seeded on 8-well ibidi™ plates coated with PureCol or on glass coverslips in an MW12 plate (uncoated) and fixed after transfection with 4% PFA at room temperature for 10 min following 2 washes with 1X PBS PFA action, which was halted by a 15 min incubation with 1mM Glycin. When the anti-CFTR primary antibody 570 (UNC, labeling the R cytosolic epitope of CFTR) was used, cells were permeabilized (10 min) with Triton-X 0.1%. When the anti-CFTR primary antibody TJA9 (UNC, labeling the extracellular EL1 antigen in the MSD1 domain of CFTR) was used, cells were not permeabilized. Cells were then blocked for 30 min at room temperature with 0.1% BSA and incubated overnight at 4 °C with the primary antibody under constant oscillation to guarantee a homogeneous distribution. The secondary antibody anti-mouse mAb Alexa 488 (Abcam, Cambridge, UK) diluted 1:500 or Cy-3 conjugated, diluted 1:500 was incubated for 90 min at room temperature on constant oscillation. DAPI staining was performed after 3 washes with 1× PBS by using the Ibidi™ Mounting Medium with DAPI and cells were observed 10 min after staining. The cells were considered positive for CFTR only if the protein was detected at the plasma membrane. The experiments were performed at least twice, with two technical replicates each. For the experiments with the REPAIRv2 system and the mxABE system, at least 80 cells were pre-sorted and counted in each technical and biological replicates; for the experiments with the mxABE system, at least 70 cells were counted. The data were normalized and expressed as a percentage.

### 4.7. Cell Sorting

At least 3 × 10^6^ cells were sorted using the BD FACS Aria III instrument (Becton Dickinson, Franklin Lakes, NJ, USA) to enrich the samples with GFP-positive cells. Cells were trypsinized and centrifuged (1000 rpm, 8 min) and then re-suspended in buffer containing 2% FBS and 1 mM EDTA in 1× PBS. The final concentration of the cells was 1 × 10^6^/mL. Cells were then sorted in MEM supplemented with ciprofloxacin and pelleted for RNA extraction or re-seeded for IF experiments.

### 4.8. Statistical Analysis

All the quantitative data in the graphs are shown as means of at least 2 independent experiments with a minimum of 2 internal technical replicates, with the error bars representing the standard error of the mean (SEM). The significance was tested using a two-tailed Student’s *t*-test with Prism (GraphPad software 7.0). *p* values are indicated as * *p* ≤ 0.05, ** *p* < 0.01.

## Figures and Tables

**Figure 1 ijms-24-10940-f001:**
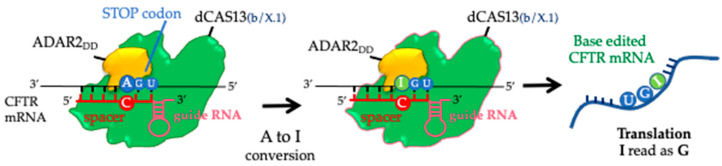
Schematic representation of RNA editing by dCas13-ADAR2_DD_ fusion proteins.

**Figure 2 ijms-24-10940-f002:**
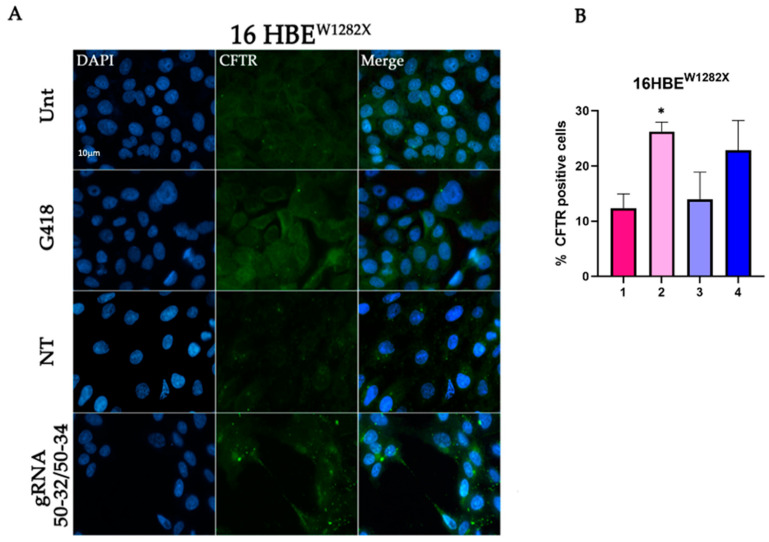
Immunofluorescence detection of CFTR protein in 16HBE^W1282X^ cells transfected with REPAIRv2 and quantification of CFTR-positive cells. (**A**) Representative fluorescence microscopy images displaying CFTR recovery. Untransfected cells (Unt, negative control), transfected cells with the plasmids encoding the indicated gRNA and Cas13b/ADAR2DD, and cells transfected with a non-targeting (NT) are shown. Cells treated with G418 were used as a positive control. Nuclei (blue) were DAPI-stained. Images were taken at 63× magnification on a ZEISS microscope equipped for epifluorescence. (**B**) Quantification of the fluorescence levels showing rescue of CFTR protein 1: 16HBE^W1282X^ Unt; 2: 16HBE^W1282X^ + G418; 3: 16HBE^W1282X^ + mxABE-NT; 4: 16HBE^W1282X^ + mxABE-gRNA. The graph shows the mean of two independent experiments with two internal technical replicates each. The error bars represent the standard error of the mean (SEM). * *p* ≤ 0.05.

**Figure 3 ijms-24-10940-f003:**
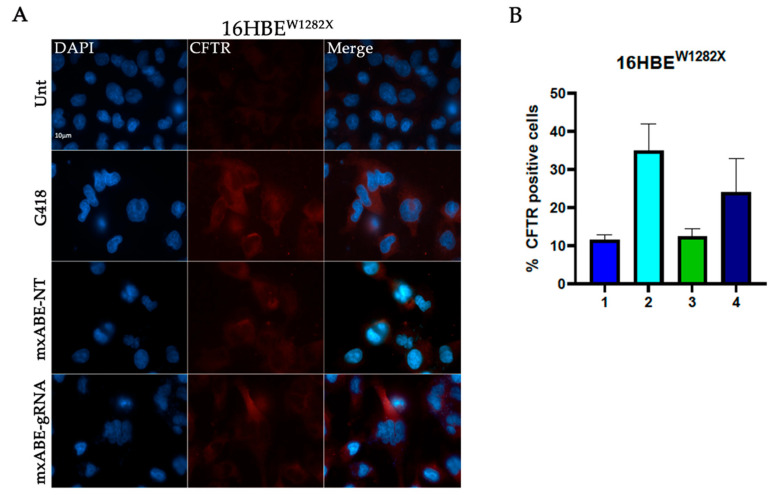
Immunofluorescence detection of CFTR protein in 16HBE^W1282X^cells following transfection with the mxABE tool. (**A**) Representative fluorescence microscopy images displaying CFTR signal (red) in cells transfected with the mxABE vector. Untransfected cells (Unt, negative control), transfected cells with the mxABE vector encoding the non-targeting (NT), and the specific gRNA for the W1282X mutation are shown. Cells treated with G418 were used as a positive control. Nuclei (blue) were DAPI-stained. Images were taken at 63×–100× magnification on a ZEISS microscope equipped for epifluorescence. (**B**) Quantification of the CFTR-positive cells according to the total cells analyzed. 1. 16HBE^W1282X^ Untreated; 2. 16HBE^W1282X^ + G418; 3. 16HBE^W1282X^ + mxABE-NT; 4. 16HBE^W1282X^ + mxABE-gRNA. The graph shows the mean of two independent experiments with two internal technical replicates each. The error bars represent the standard error of the mean (SEM).

**Figure 4 ijms-24-10940-f004:**
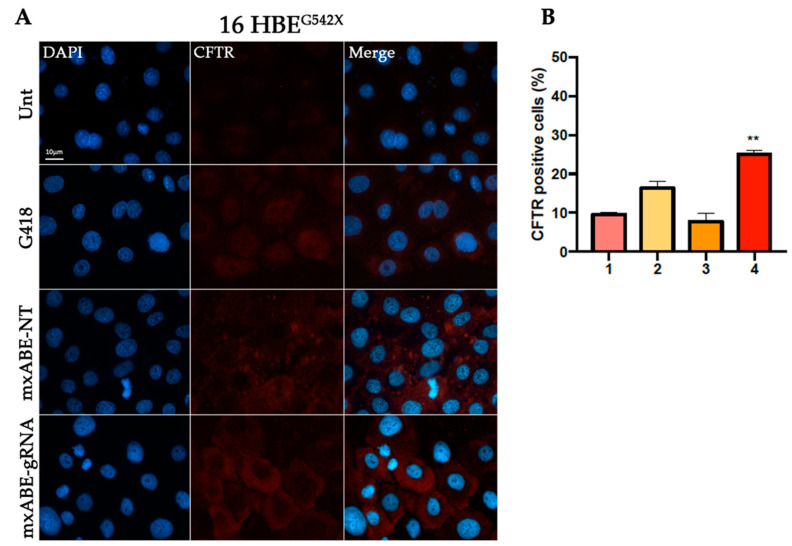
Immunofluorescence detection of CFTR protein in 16HBE^G542X^ cells following transfection. (**A**) Representative fluorescence microscopy images displaying CFTR signal (red) in cells transfected with the mxABE vector. Untransfected cells (Unt, negative control), transfected cells with the mxABE vector encoding the non-targeting (NT), and the specific gRNA for the G542X mutation are shown. Cells treated with G418 were used as a positive control. Nuclei (blue) were DAPI-stained. Images were taken at 63x magnification on a ZEISS microscope equipped for epifluorescence. (**B**) Percentage of the CFTR-positive cells according to the total of cells analyzed. 1. 16HBE ^G542X^. Untreated; 2. 16 HBE^G542X^ + G418; 3. 16 HBE ^G542X^ + mxABE-NT; 4. 16 HBE ^G542X^ + mxABE-gRNA. The graph shows the mean of two independent experiments with two internal technical replicates. The error bars represent the standard error of the mean (SEM). ** *p* ≤ 0.01.

**Figure 5 ijms-24-10940-f005:**
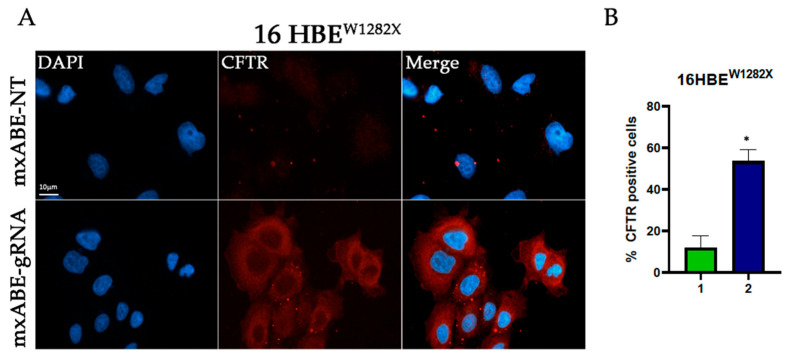
Immunofluorescence assay on sorted 16 HBE^W1282X^ cells. (**A**) Representative images showing the recovery of the CFTR protein (red) after transfection with the mxABE vector containing the specific gRNA for the W1282X mutation. The CFTR protein can also be observed at the cell membrane. (**B**) Percentage of the CFTR-positive cells according to the total of the cells analyzed. 1. 16HBE ^W1282X^ + mxABE-NT; 2. 16HBE ^W1282X^ + mxABE-gRNA. The graph shows the mean of two independent experiments with two internal technical replicates each. The error bars represent the standard error of the mean (SEM). * *p* ≤ 0.05.

**Figure 6 ijms-24-10940-f006:**
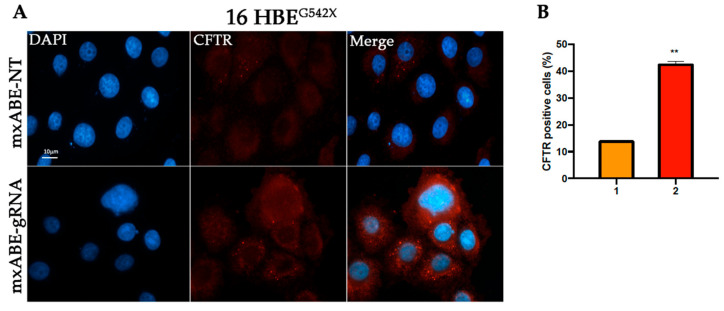
Immunofluorescence assay on sorted 16HBE^G542X^ cells. (**A**) Representative images showing the recovery of the CFTR protein (red) after transfection with the mxABE vector containing the specific gRNA for the G542X mutation. The CFTR protein can also be observed at the cell membrane. (**B**) Percentage of the CFTR-positive cells according to the total cells analyzed. 1. 16HBE^G542X^ + mxABE-NT; 2. 16HBE^G542X^ + mxABE-gRNA. The graph shows the mean of two independent experiments with two internal technical replicates. The error bars represent the standard error of the mean (SEM). ** *p* ≤ 0.01.

**Figure 7 ijms-24-10940-f007:**
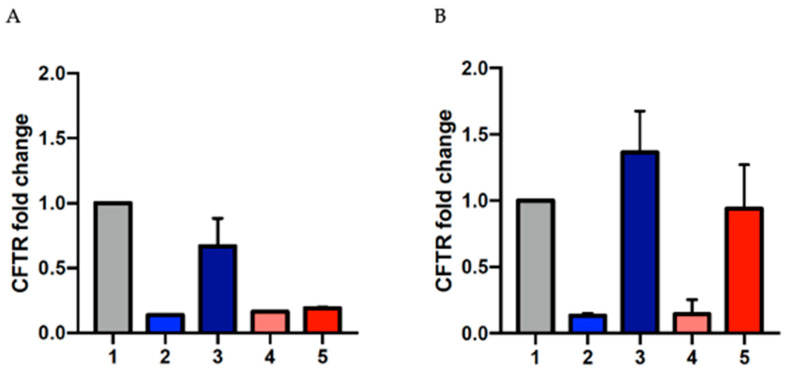
Increased CFTR transcript levels following transfection. RT-qPCR histograms showing the increased amount of CFTR mRNA in 16 HBE^W1282X^ and HBE^G542X^ cells after treatment with the specific mxABE plasmids in respect to the WT, before (**A**) and after (**B**) cell sorting; 1: 16HBE WT, 2: 16HBE^W1282X^ untransfected, 3: 16HBE^W1282X^ mxABE transfected, 4: 16HBE^G542X^ untransfected, 5: 16HBE^G542X^ mxABE transfected. The graphs show the mean of two replicates. The error bars represent the standard error of the mean (SEM).

## Data Availability

The data presented in this study are available here and in Appendix A.

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
