# Peer review of "Site-Specific RNA Editing of Stop Mutations in the CFTR mRNA of Human Bronchial Cultured Cells"

_ijms, 2023, doi:10.3390/ijms241310940_

Round 1

Reviewer 1 Report

Thank you for giving me the opportunity to review this very interesting work.

The authors investigated the possibility of rescuing the CFTR nonsense mutation (UGA) by sequence-specific RNA editing in CFTR mutant CFF-16HBEge W1282X and G542X human bronchial cells. They used two different base editor tools that take advantage of ADAR enzymes (adenosine deaminase acting on RNA) to edit Adenosine to Inosine (A-to-I) within the mRNA: the REPAIRv2 (RNA Editing for Programmable A to I Replacement, version2) and the minixABE (A to I Base Editor).

  • When abbreviations are used, spell out the full word at first mention in the text followed by the abbreviation in the parentheses. Thereafter, use the abbreviation throughout.
  • Have the authors tried using primary cells directly from patients? For example, have the authors tried to carry out the same experiments in epithelial nasal cells directly collected from Cystic Fibrosis patients?
  • The authors do not mention statistics in the materials and methods. Have you used it? if yes, you should also mention it in figures and in Materials and methods

Reviewer 2 Report

The author used two previously published approaches (REPAIRv2 and mxABE) to sequence-specifically edit RNA in order to recover the CFTR protein. The article is well-written, but the innovation part is not distinct. The authors didn't develop new approaches; rather, they applied the two existing approaches to different cells.

Major revision is suggested. 

1. There are some missing citations in the introduction, specifically in lines 34-36, 46-51, and 59-67.

2. Figure 1 is unclear and should be reconsidered for clarity.

3. Scale bars are missing in Figure 2, Figure 3, and Figure 5. To improve clarity for readers, please consider adding them.

4. Error bars and statistical analysis are not provided for the data presented in this manuscript. Please include this information.

5. The author referred to two CRISPR-Cas13a-based approaches for RNA editing. One major concern with the CRISPR technique is its off-target effects. Did the authors encounter this issue in their work? It would be valuable for the authors to discuss how they plan to overcome it if the approach used in this work is intended for therapeutic use in the future.

Reviewer 3 Report

In their manuscript entitled "Site-specific RNA editing of stop mutations in the CFTR mRNA of human bronchial cultured cells", Chiavetta and co-authors describe the use of two different tools that take advantage of adenosine deaminase activity to edit Adenosine to Inosine (A-to-I) changes within the mRNA, the REPAIRv2 and the minixABE, to repair the mRNA of CFTR mutants with stop codons that do not allow the formation of the complete protein. The authors report the required selection of guiding sequences, cloning and transfection of cell lines. Immunofluorescence experiments are reported for cells transfected with both RNA editing tools, shoing tha tboth were able to recover the CFTR protein in the CFTR mutant cells. Authors also used  RT-qPCR to confirm the rescue of the CFTR full transcript. The manuscript is well organized and written. 

The topic is worth to study and the authors have used cultured epithelial cell lines to investigate the feasibility of mRNA editing to correct the genetic defect associated with about 10% of the CFTR mutations, premature stop codons. If the methodology works on a living being is still far from being demonstrated. There are some issues the authors should address:

1) throught the text , the system minixABE is often referred also as mini xABE. Only one designation should be used.

2) In the abstract is "CFF-16HBEge" correct?

3) line 41: ADARs 1/2 ?

4) line 48: aminoacid?

5) line 97: "Cells treated with G418 were used as a positive control." This is the firs ttime G418 appears and no explanation is given why use this compound. The methods only mention the compound but nothing is mentioned on why use this compound.

6) some abbreviations appear without any definition,  as for instance RRID.

7) Lines 254-262: rewrite the whole section, as the use of ":" and ", "  is confusing, and the whole sentence is hard to understand. The last part"..., mxABE-NT, mxABE-261 CFW1282X 50-25; mxABE-CFG542X 50-25" means what? Correct the word "vectore".

Round 2

Reviewer 2 Report

The author has addressed every comment. Acceptance is suggested.